

# Evaluating the suitability of documents on the prevention of major risks intended for the general public

Laetitia Ferrer[1], Corinne Curt[1]

[1]INRAE, Aix Marseille Univ, RECOVER, 3275 Route Cézanne, CS 40061, 13182 Aix-en-Provence, France

*Correspondence to*: Corinne Curt (corinne.curt@inrae.fr)

**Abstract.** Providing preventive information to the public is one of the major challenges in the prevention of major hazards. In France, the DICRIM (Document d'Information Communal sur les Risques Majeurs; in English "Municipal Information Document on Major Risks") is currently the main regulatory risk communication tool intended for the general public. It has been developed with the aim of contributing to awareness and knowledge of risks and informing on the actions and behaviors to adopt. This raises questions as to its suitability composed of readability and legibility, essential for its appropriation by the population. Answering these questions constitutes the first step in improving its effectiveness. This article presents the development of indicators and a decision support model used to analyze and improve the suitability of a DICRIM. Two types of indicator, considering visual and content characteristics, were developed: System Indicators for the entire document and Substantive and Formal Component Indicators for the fields it contains. They were formalized using five types of source: the literature, expert knowledge, opinions of residents, analysis of a DICRIM database and a questionnaire survey. The aggregation of indicators provides suitability scores and feedback on what should be improved in the DICRIM. Validation sessions were performed with five risk analysis experts. The advantage of the model is that it can be used by the town halls and design offices of any municipality without the need to call on experts or significant human or financial resources.

## 1 Introduction

### 1.1 The importance of risk communication to raise public awareness and preparedness

Each year in the world, major natural and technological phenomena cause disasters with considerable damage to human life and property. There is a broad agreement on the importance of raising public awareness and preparedness by effective communication (Guzzo et al., 2014; Höppner et al., 2012). Risk communication can be viewed as "a process for facilitating a better and broader understanding of the risks that people face, while – importantly – also improving their decision-making capabilities in associated risk management contexts" (Árvai, 2014). It aims at informing, persuading and facilitating public support for hazard risk mitigation and preparedness (Sanquini et al., 2016). The goal is to reach the highest possible number of people at risk (Maidl and Buchecker, 2015) and inform and help individuals make an informed decision.





The awareness about risk communication is a global issue. For the United Nations, the public must be informed periodically
of the dangers and levels of risk to which they are exposed, and how this may be changing (United Nations, 2006). The aim of
Global Target G of the Sendai Framework for Disaster Risk Reduction is to "substantially increase the availability of and
access to […] disaster risk information and assessments to people by 2030" (United Nations/International Strategy for Disaster
Reduction, 2015). Around the world, numerous initiatives have been developed to improve risk communication aimed at the
general public, e.g., the National Centre for Environmental Data and Surveillance in England, government campaigns in the
Netherlands (www.crisis.nl), and communication as an integral part of organizational risk management in Austria, New
Zealand and Canada. In France, since October 1990, the law has required local authorities to implement various communication
measures (municipal documents, posters, meetings, flood markers, etc.) to enable citizens to be aware and informed of the
major risks to which they may be exposed (Vigier et al., 2019).

Risk communication is recognized as one of the most common process to build social capacity (Höppner et al., 2010; Maidl
et al., 2021), improve awareness (Höppner et al., 2010; Guzzo et al., 2014) and risk perception (Kim and Madison, 2020),
develop a sense of self-responsibility, encourage precautionary measures and search for further information sources
(Hagemeier-Klose and Wagner, 2009; Höppner et al., 2010; Guzzo et al., 2014; Kievik and Gutteling, 2011), change people's
attitude and move toward more disaster resilient communities (Steelman and Mccaffrey, 2013; Höppner et al., 2010; Lindell
and Perry, 2004). The basic assumption is that providing relevant information would enable the people to identify the hazardous
phenomenon and its features and motivate them to change their behavior, lead them to make relevant decisions (Maidl et al.,
2021; Demeritt and Nobert, 2014; Maidl and Buchecker, 2015).

Models have been proposed in the literature to characterize the people's response during environmental hazards and disasters.
The Protective Action Decision Model (PADM) developed by (Lindell and Perry, 2012) identifies three critical predecision
processes – reception, attention, and comprehension (of warnings or exposure, attention, and interpretation of
environmental/social cues) – completed by protective action decision making. Ripberger et al. (2019) proposed three criteria:
forecast and warning reception, comprehension, and response. These processes/criteria can be adapted to the risk preparedness
phase: the reception and comprehension of preventive materials and the response following its analysis constitute the main
objectives of the risk communication in this phase (Figure 1).

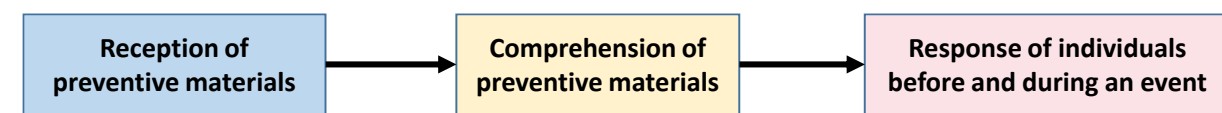


**Figure 1. Main objectives of the risk communication during the preparedness phase**

Possible cognitive biases may occur when an individual receives preventive information (illusion of control (Langer, 1975),
optimism bias (Kahneman, 2011), attitude of denial (Weiss et al., 2006), etc.), impeding understanding of its content. However,
"even though good risk communication cannot always be expected to improve a situation, poor risk communication will nearly



always make it worse" (National Research Council, 1989). Therefore, the communication of information remains a major factor in preventing population exposure to risk. Indeed, it can constitute a form of indirect risk experience and in this sense a means of strengthening its acceptance and the individual involvement of exposed populations (Lindell and Perry, 2004). Some shortcomings are identified in the literature: uneven implementation and lack of control of the resources (Irma et al., 2015),

different behavioral instructions from one document to another, even with reference to the same phenomenon (Bellurot et al., 2014), incomplete information on the dangers to which the population is exposed, etc. Therefore, communication effectiveness constitutes a main issue.

## 1.2 Effective risk management requires effective communication

Communication effectiveness is defined as "the extent to which the intended goals of communicative actions are realized"

(Boholm, 2019b): a communication is intended to be helpful and obtain the expected impact. "Risk communication is successful to the extent that it raises the level of understanding of relevant issues or actions" (National Research Council, 1989). Effective risk management requires communication that spans all successive phases of risk management, before, during and after events (Lindell and Perry, 2004): the effectiveness of communication during an event also depends on the effectiveness of prior communication that better prepares people for future events (Höppner et al., 2010). Achieving effective

communication means that understandable ways of presenting risk, a complex concept tainted with uncertainty, must be found (Slovic, 1986).

Effectiveness concerns the three main objectives described in Figure 1. If information is efficiently transmitted, people will have better knowledge of hazards and safety rules and will be prompted to make protective action decision (Siegrist and Cvetkovich, 2000; Terpstra et al., 2009).

In this article we focus on the comprehension phase (Figure 2); the preventive material is supposed received by the recipients. Moreover, we consider that the preventive material is a document. Our work relies on the following assumption: if the document is suitable *i.e.* appropriately framed (content) and visually compelling (format), then people will be likely to read it. If people read the document and understand clearly the information presented then the message would participate to take preventive actions and adequate actions during an event ("Response" in Figure 1). This is in line with results from Siegrist and

Cvetkovich (2000), Lindell and Perry (2004), Höppner et al. (2012), Covi and Kain (2016) and Boholm (2019a). So, improved comprehension is expected through adequate suitability composed of readability and legibility (Figure 2). Readability is defined as the "the ease of understanding or comprehension due to the style of writing" (Rooney et al., 2020). Legibility concerns typeface (such as serifs, font sizes, weight, contrast, etc.) and layout (Dubay, 2004). It considers the visual properties of the text, which are critical for an efficient readability (Neuhauser et al., 2013). Using a precise and concrete language,

showing pictures and graphs, or giving individual case histories participate to attract and hold people attention (Dransch et al., 2010). In the following, we will use the terms "format" and "content".



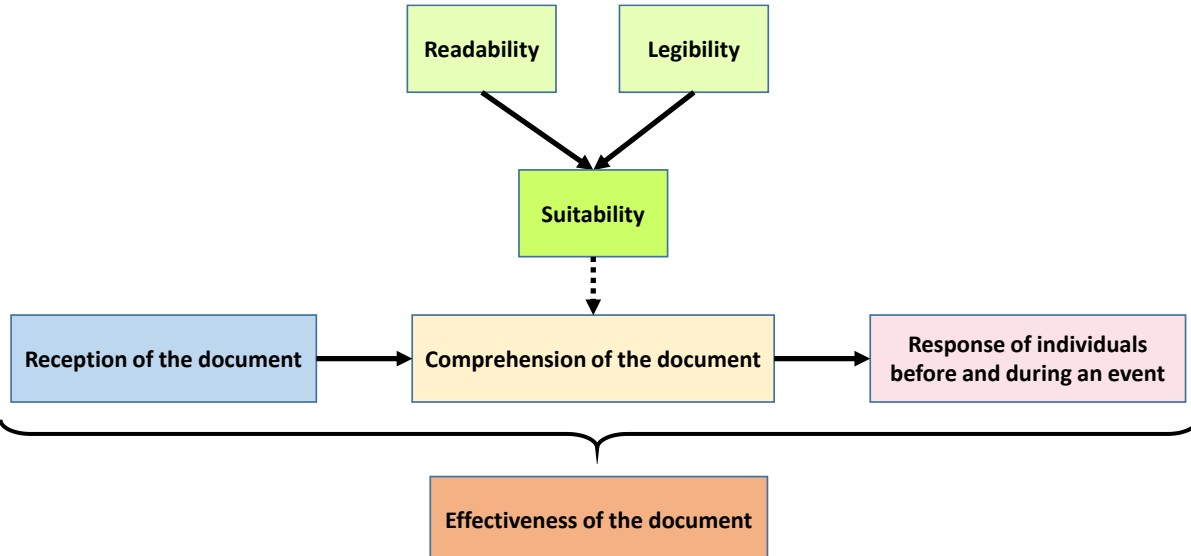

**Figure 2. Risk communication effectiveness depends on reception, comprehension and response processes**


### 1.3 Only a few works have dealt with the suitability of risk management materials

Only few works dealt with the suitability of risk management materials (Neuhauser et al., 2013; Höppner et al., 2010). Numerous readability formulas are reported in the literature, however they mostly rely on two criteria (Doak et al., 1996), words difficulty and sentence length. For instance, the Flesch Reading Ease test and the Flesch–Kincaid Grade, probably the most commonly used tests, were used to determine the comprehension difficulty of transcribed interviews concerning the ecological risk assessment of pesticides (Hunka et al., 2013). However, these formulas are not sufficient to assess message format and content. Besides them, the Suitability Assessment of Materials (SAM) is the most validated and most commonly used method (Doak et al., 1996) that assesses readability and legibility. It stems from the health area and remains mainly used in this area although used in two studies related to risk communication (Friedman et al., 2008; Neuhauser et al., 2013). The SAM method is much more elaborated than readability formulas. It assesses 21 factors related to Content, Literacy Demand, Graphics, Layout and Typography, Learning Stimulation and Motivation, Cultural Appropriateness. However, it does not distinguish between different parts of a document (e.g. cover page, summary, information for preventive actions…) and, as elaborated for health documents, it does not analyze information materials such as design of maps, photo resolution or number of photos, etc. Other studies dealt more specifically with maps (Cao et al., 2016; Hagemeier-Klose and Wagner, 2009; Macpherson-Krutsky et al., 2020). A methodology relying on a set of indicators to evaluate the legibility and readability characteristics collected from the literature was proposed (Serna Sans et al., 2018). However, this approach is dedicated to the comprehension of informative panels on natural landscape and not to risk documents. Moreover, it does not consider elements such as the document format, features of the photos or colors used in the document for instance.





### 1.4 Aim of the article

From this perspective, the aim of our work is to propose a method for evaluating the suitability in terms of form and content, of documents on the prevention of major risks intended for the general public and decision-making tools for the stakeholders responsible for their implementation. Our method is based on indicators specific to the risk issue which allow the assessment of various parts of the document and information materials. We develop two tools to implement the method. The method and tools could be used generically, regardless of the municipality and the risks involved, reducing the resources to be mobilized.

We chose to implement our methodology to the DICRIM (in English "Municipal Information Document on Major Risks") because it is the main reference document in terms of informing the public about the natural and technological risks affecting the municipal territory in the French government's overall strategy for risk prevention.

## 2. Development of a method for assessing the suitability of documents on the prevention of risks

### 2.1 Implementation case: the DICRIM

The DICRIM concerns all the 11 major risks that may affect a municipality (floods, earthquakes, ground movements, forest fires, avalanches, storms/cyclones, volcanic eruptions, dam failures, nuclear accidents, industrial accidents and mining risks) and recommends actions in case of an event. The mayor establishes the DICRIM. In paper form it can be consulted at the town hall by the public. In some cities, it is also distributed directly to residents' mailboxes or posted on the municipal website in digital format or, more rarely, made available in interactive mode.

The main headings and information to be included in the DICRIM are defined in the National Model for the Application of the Environment Code (articles L 125 - 2 and R 125 - 9 to R 125 - 27) issued by the MTES (Ministry of Solidarity and Ecological Transition) (Medde, 2013). However, this model provides only very general recommendations on the background of the DICRIM and no standards exist for its visual content. Following the requirements described in Medde (2013) the DICRIM is composed of an editorial from the mayor, a DICRIM presentation, substantial information for each hazard affecting

the town (risk presentation, prevention and protection actions, safety instructions, mapping), information about other preventive informational methods (public notices, floodmarks, etc.), emergency phone numbers and equipment to always have at home to be ready. This leads to disparate documents in terms of content and form that vary from one municipality to another. Research work has already been carried out on the DICRIM (Bellurot et al., 2014; Douvinet et al., 2013; Gominet, 2007). However, the question of a robust method for evaluating its suitability remains. The absence of a formal and detailed

framework for its form and the lack of public familiarity make its suitability difficult to assess without a rational approach.

### 2.2 Proposed approach

Figure 3 shows the approach used to evaluate the suitability of documents dedicated to risk prevention and notably DICRIMs. The different steps are detailed in the following paragraphs.





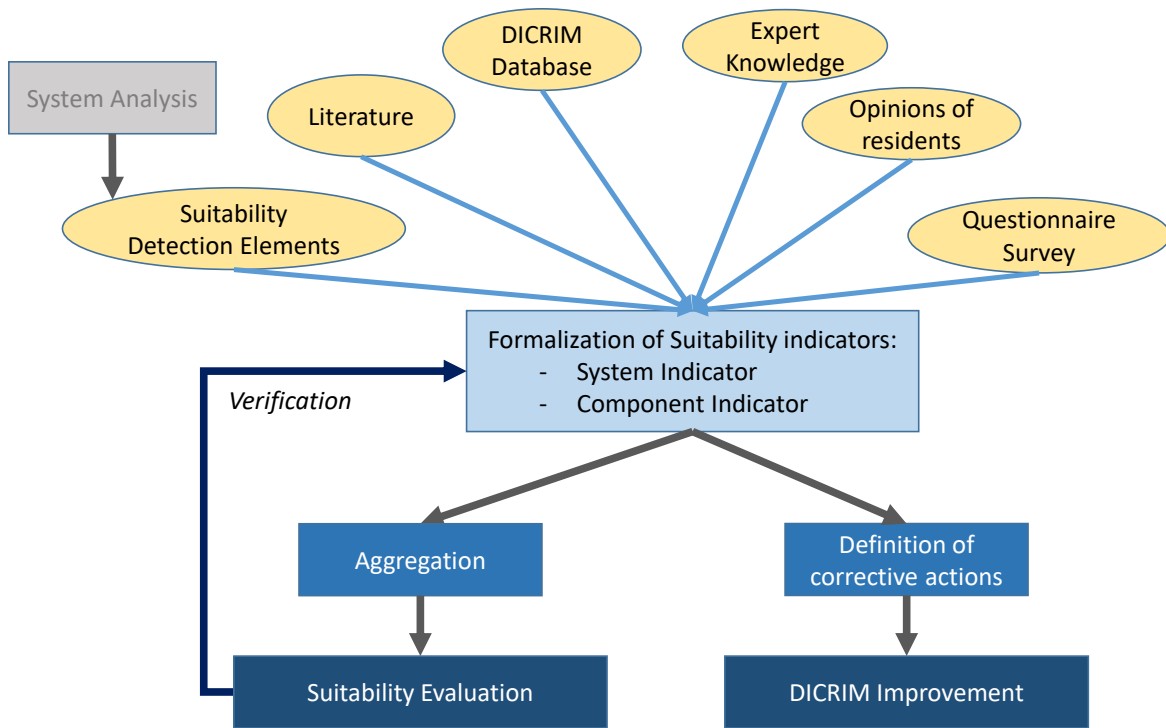

**Figure 3: Proposed approach for the construction of a model for assessing the suitability of a DICRIM**

## 2.3 Two scales of analysis

The evaluation is carried out on two scales: the entire document (System Indicator in Figure 3) and the document's components (content and form). These two granularities were considered because they are generally the two phases of discovery of a document by the reader. First, the document is analyzed superficially (first scale) before being opened and looked at in more detail (second scale). The first step is also important to consider because the excessive inefficiency of the document could block continued reading and thus lead to the abandonment of the second step. Each suitability detection element (stemming from the systemic analysis (Ferrer et al., 2018)) is then assigned to one or more components and formalized as a suitability indicator. The same detection element could therefore have generated several indicators. There are therefore System Indicators coded SIx for the scale of the entire document and Component Indicators coded CIx for the scale of the document's components.



## 2.4 Definition of suitability detection elements

As the works found in the literature are not able to fully analyze risk prevention documents, we first analyzed the system under
study using a systemic analysis method based on a structural, functional and dysfunctional analysis. It can be used to analyze
the content of a document dedicated to risk communication in a generic way (Ferrer et al., 2018). An application was performed
for the DICRIM. The systemic analysis was implemented at the system level (Document) and at the level of its components
(for instance, the summary, the components dealing with risks, emergency telephone numbers, etc.). The outputs of this step
are suitability elements at the system level on the one hand and at the component level on the other. It considers in particular
the form and content of documents not taken into account in the regulations. These include, for example, "Number of photos"
and "Presence of catch phrases". The suitability detection elements must be grouped to facilitate the assessment and make it
robust.

## 2.5 Indicators formalization

We build the indicators according to a formalization grid existing in the literature (Curt et al., 2010; Talon et al., 2014) that
allows obtaining the information necessary to correctly use the indicators: repeatability and reproducibility must be achieved.
The indicator grid is made up of a definition, possibly completed with operating conditions, a measurement scale with
anchorage points (photographs, diagrams, and linguistic descriptions), spatial characteristics (sampling and measurement
location), and time characteristics (measurement frequency, analysis frequency, etc.). Various choices of scales are possible
and have been discussed in the work of Curt et al. (2010): nominal, ordinal, interval, ratio scales. We use a single identical
scale for each of the indicators for our evaluations and propose using a double assessment scale for all the indicators: an ordinal
scale (ranging from 0/Unacceptable to 10/Good) characterized by ordering categories as a function of an intensity criterion
linked to a scale of intervals that permits working on continuous numerical magnitudes (Table 1). For example, if the format
of the document is A4 (resp. A5) format, then the assigned score is 5 (resp. 10).

| Name | SI1 – DICRIM format |
|---|---|
| Definition | Efforts are being made to evaluate the format of the DICRIM when it is received in booklet form. |
| Scale and references |  10: A5 (14.8 cm x 21 cm) 5: A4 (21 cm x 29.7 cm) 3: A6 (10.5 cm x 14.8 cm) |
| Location characteristic | Global DICRIM |

**Table 1: System Indicator 1 "DICRIM Format" Grid**



The choice of a single scale requires particular vigilance regarding the consistency between the references associated with the scale of the indicators. References are associated with the assessment scale for each indicator: they provide the anchoring points on the scale and allow describing the different possible statuses of the indicator. These references are specific to each

indicator. Each situation that may be encountered corresponds to a reference. The interest of using the same scale is twofold: the risk of scoring errors due to confusing scales is avoided and its use simplifies subsequent combinations because it allows, in certain cases, combinations of type "minimum", "maximum", sum, etc. through simple arithmetic.

To formalize the suitability indicators, *i.e.* provide the definition and define the milestones of the scale (anchorage points), various sources may be used: the literature, expert knowledge, opinions of residents, analysis of databases, survey, etc. The

literature in the fields of Communication Sciences, Cartography, Risk Geography, Sociology, Cognitive Sciences, etc. allows identifying existing recommendations to promote effective communication.

In our study, five sources were used to formalize the suitability indicators: the literature e.g. Doak et al. (1996), Dransch et al. (2010), Dubay (2004), Hagemeier-Klose and Wagner (2009), Macpherson-Krutsky et al. (2020), Serna Sans et al. (2018); expert knowledge; opinions of residents obtained from an exploratory survey; analysis of a DICRIM database; and finally a

questionnaire survey for a specific analysis on the first page of the document. These are complementary ways of collecting the information necessary for the indicator definition and references on the assessment scale.

Three experts in communication, decision support and risk prevention were interviewed to gather information to create the milestones for the indicators. Two are experts who have performed research in technological or natural risk analysis for more than 15 years, while the third is a communication expert with 27 years' experience in the field. In parallel, elements of residents'

speech were used from an exploratory survey conducted on a multi-risk municipality.

The non-directive interviews were conducted to identify the positive and negative opinions of 10 respondents about the DICRIM and suggestions for improvement. The point of view of the general public must be taken into account carefully because they are the future users of the DICRIM. Several of these views were consistent with expert opinions or the literature, and elements not previously mentioned made it possible to refine the indicators description. These included, for example, the

total number of pages or sections recommended for the document, and useful data to be mentioned in the form of more definitions or suggestions for more attractive visuals.

In the event of missing information from the various sources, the use of a database composed of about fifty DICRIMs from all over France and with various characteristics (number of pages, presentation format, quantity of information contained, etc.), made extrapolation possible in order to construct definitions and scales.

Finally, we decided to conduct a questionnaire survey specifically dedicated to the design of the cover page. Indeed, some of the document's characteristics were difficult to formalize because of the subjectivity involved in their analyses as diverse styles of presentation are possible: visuals that are figurative, humorous, modern, refined, simple, inspire a feeling of fear, etc. No recommendations or requirements exist in the law for this aspect, which leads to considerable diversity in the form and content of the DICRIM in France. In order to build these indicators, we needed to get an idea of the visual characteristics that may or





may not have an impact on the reader and why. The questionnaire was conducted with the staff of the INRAE (French National Research Institute for Agriculture, Food and the Environment) Aix-en-Provence site. Respondents can be asked to choose the three pages, among nine proposed, that appeal most to them in order of preference and, conversely, those that least appeal to them. The emotion felt when viewing the different pages chosen can also be elicited. The results allowed us to construct the CI1 indicator grid "Presentation Style" for the Cover Page Component.

When constructing the indicator grids, particular attention was paid to the vocabulary used. The indicator grids must be usable by all the municipal services or design offices without the need for expert intervention.

## 2.6 Assessment and improvement models

A model is built based on the formalized indicators (Figure 3). It provides a detailed analysis, with an assessment of the suitability of the form and content of the document. The indicators are currently aggregated by a weighted average. The output
data is composed of the scores assigned by each indicator, the final suitability score, and any feedback to be provided to improve the document. To remedy an indicator score considered insufficient (from a score of 0 to a score of 5 included, for example), the model proposes the actions to be implemented at the component or system level. These improvements will be made so that the indicator reaches a score of 10 on a new evaluation, ensuring better suitability of the given component or system. For instance, if a DICRIM has an A4 format (score assigned equals 5) (Table 1), then the feedback model proposes to
present the document as an A5 format (score assigned equals 10).

## 2.7 Verification

A verification phase is carried out on all the indicators with experts following the approach provided in Curt et al. (2010). The objective is to check on the content of the grids, as well as their implementation (e.g. verification of the evaluation duration). The aim of this step is to verify the content of the grids. If a difference exists between scores given by several experts, this
may stem in particular from the formulation of milestones that are too ambiguous or complex. If such differences are noted during the verification sessions, then the references on scale are altered to improve the corresponding grids. Independently, experts assess several existing DICRIMs in real life situations of grid use. This allows exploring different references in each indicator grid. At the same time, they have to note any difficulties they might have encountered or suggestions for changes they had to make to a particular grid. These verifications make it possible to highlight improvements that have to be made to
certain indicators and to justify the robustness of the evaluation.

To interpret the results, statistical standard deviation calculation treatments are performed to analyze rating deviations, mean to aggregate ratings and an ANOVA to assess whether the differences between assessors are significant.

Verification sessions of the indicators were conducted with five risk analysis experts. These experts were different from those asked to build the indicators. Two of them are risk analysis engineers with respectively 25 and 5 years' experience, the third
has performed research in technological and natural risk analysis for more than 15 years, and the last two are junior experts pursuing a 5-year training course on natural disaster risk management and planning.



The experts evaluated three existing DICRIMs in real life situations of grid use. The DCIRIMS assessed were different in terms of length, visual aspect and contained information. This allowed exploring different references in each indicator grid. Each expert was provided with a single file containing the following: the test protocol; the grids of system indicators and

indicators for the components; three existing DICRIMs (DICRIM 1 (D1) =28 pages, 9 risks, modern and dense visuals, DICRIM 2 (D2) =9 pages, 9 risks, aged and aerated visuals; DICRIM 3 (D3) =32 pages, 7 risks, old and dense visuals); the rating scale; the Excel file in which the experts had to enter their scores. The experts had to conduct the evaluation of the three DICRIMs using the indicator grids and time themselves for each evaluation. At the same time, they had to note any difficulties they might have encountered or suggestions for changes they had to make to a particular grid.

## 3. Indicators for evaluating DICRIM suitability

### 3.1 Formalization of system indicators dedicated to the entire DICRIM

To analyze the suitability of the document as a whole, 13 detection elements were identified following the systemic analysis (Ferrer et al., 2018). These 13 elements were directly formalized in the form of indicators: DICRIM format, Visual printing (color or black and white), Total number of DICRIM pages, Evaluation of added components, Evaluation of the added

component "multi-risk map", Evaluation of the added component "Information on the insurance system", Typography of DICRIM texts, etc. By way of example, Table 1 presents System Indicator 1 (SI1).

As seen above, the law imposes a number of components that must be included in the DICRIM and which serve as a basis for some of our analyses. However, the mayors are free to add headings in their DICRIM if they consider them relevant for transmitting preventive information. The SI9 indicator is used to evaluate the main added items frequently encountered in the

DICRIMs of our database. This regularly involves the addition of a multi-risk map that presents all the hazards of the municipality concerned on the same map. This has been identified as a particularly relevant tool for the communication on multi-risk (Curt, 2021). As this map has no regulatory specifications, we developed SI11 to assess its suitability (cf. Table 2).


| Name | SI11 – Evaluation of the added "multi-risk map" component. |
|---|---|
| Definition | If a map with several risks is presented, this indicator is used to assess its legibility. Please complete it only if this map is present. |
| Scale and references | |

<table>
<tr><td>Unacceptable<br>0   1</td><td>Bad<br>2   3</td><td>Poor<br>4   5</td><td>Acceptable<br>6   7</td><td>Good<br>8   9   10</td></tr>
</table>

10: The map has:

- a scale fine enough for the observer to locate points of interest while encompassing the entire city;

- an adapted semiology so that the extension areas of each risk can be identified;

- colors for each risk identical to those used on each of the maps dedicated to a single risk present in the risk components;

- sharpness allowing a clear legibility;

- a scale clearly informing the observer about the meaning of each figure on the map;

- information conveyed that can be understood by the largest number of people;

7: one of the above characteristics are not provided;

5: two of the above characteristics are not provided;

2: more than two of the above characteristics are not provided.

| Location characteristic | Global DICRIM |
|---|---|

Table 2: System indicator 10 grid "Evaluation of the added component "multi-risk maps""


## 3.2 Formalization of content and form indicators dedicated to DICRIM components

Nine components have been identified: the cover page (Cp1), the editorial (Cp2), the summary (Cp3), the presentation of the DICRIM and the prevention of major risks in France (Cp4), the risk component(s) (depending on the number of risks present in the given municipality) (Cp5), the municipal poster (Cp6), "where to get more information" (Cp7), emergency telephone

numbers (Cp8) and equipment to have at home at all times in order to be ready (Cp9).

During the systemic analysis, 14 form elements were identified: Presentation style, Element represented in the photograph, Photograph resolution, Component length, Relevance of the association of text and background colors, Number of photographs, Photograph size, Proportion between illustrations, text and whitespace, Title typography, Title colors, Text structure, Drawing type, Map type, Map size. Concerning the content,six basic elements were identified: Adaptation of the

title to the content of the section, Vocabulary, Presence of catch phrases, Summary nature of paragraphs, Usefulness of the data presented, Legibility of diagrams. However, for their formalization, it was necessary to assign each of these elements to each component, if justified, to evaluate its suitability. A total of 48 component indicators (31 related to form – 17 related to



content) have been formalized, allowing the analysis of the content and visuals of the DICRIM. The indicators have the same name as the detection elements. Table 3 shows the distribution of the detection elements for each component (cross in the table). The crosses in the table are more numerous than the total number of indicator grids developed because the same indicator can be used for several components.

| | Cp1 | Cp2 | Cp3 | Cp4 | Cp5 | Cp6 | Cp7 | Cp8 | Cp9 |
|---|---|---|---|---|---|---|---|---|---|
| Style of presentation | X | X | X | X | X | | | | |
| Element represented by the photograph | X | X | X | X | X | | | | |
| Resolution of the photograph | X | X | X | X | X | X | | | |
| Length of component | | X | | X | X | X | | | |
| Pertinence of the association of text and background colors | | X | X | X | X | X | X | X | X |
| Number of photographs | | X | X | X | X | | | | |
| Size of photograph | | X | X | | X | | | | |
| Proportion between illustrations, text and whitespace | | X | | X | X | X | | | |
| Typography of titles | | X | | X | X | X | | | |
| Color of titles | | | X | | | | | | |
| Structure of the text | | | X | X | X | | | | |
| Type of drawings | | | | | X | | X | X | X |
| Type of map | | | | | X | | | | |
| Size of map | | | | | X | | | | |
| Adaptation of title to content and heading | X | X | | X | X | | | | |
| Vocabulary | | X | | X | X | X | | | |
| Presence of catch phrases | X | X | | X | X | | | | |
| Synthetic character of paragraphs | | X | | X | X | X | | | |
| Usefulness of data presented | | X | | X | X | X | X | X | X |
| Legibility of diagrams | | | | X | X | | | | |

**Table 3: Distribution of detection elements according to components – Cp1: Cover page, Cp2: Editorial, Cp3: Summary, Cp4: Presentation of the DICRIM and the prevention of major risks in France, Cp5: Risk component(s), Cp6: Municipal poster, Cp7: "Where to get more information", Cp8: Emergency telephone numbers, Cp9: equipment to have at home at all times in order to be ready**



It should be noted that no elements are used for all the components. The element "Relevance of the association of text and background colors" is the most commonly used (seven components). Some elements are used only once, such as the "Map

Size" element in the risk component or "Title Colors" in the summary. The titles constitute the main content of the summary, which explains an indicator dedicated specifically to their color. The color of the titles of the other components is processed among other typographical characteristics in the "Title Typography" indicator. The Risk Component is the one with the largest number of elements and therefore indicators (19). Indeed, it gathers all the headings of the different phenomena and a large amount of information. Assessing all the risk headings of a DICRIM would lead to the use of too many indicators.

The same indicator used for two different components may be identical but may also differ in the constitution of its milestones because it is adapted to the component it allows evaluating. For example, Table 4 shows "Element represented by the photograph" adapted to the "Editorial" component CI10.

| Name | CI10– Element represented by the photograph |
|---|---|
| Definition | If a photograph is present, this indicator evaluates what it represents. |
| Scale and references | ![Scale: Unacceptable (0–1, black), Bad (2–3, red), Poor (4–5, orange), Acceptable (6–7, yellow), Good (8–10, green)]<br><br>10: photograph of the city and/or issues and/or protection measures;<br><br>6: photograph of damage and/or phenomenon and/or mayor, other;<br><br>5: photographs of the 2 types. |
| Location characteristic | At the beginning of the DICRIM – on the page where the editorial is located. |

**Table 4: Grid of the component indicator of form 10 "Element represented by the photograph" for the Editorial component**


Table 5 shows an example of another indicator "element represented by the photograph" (CI34) but adapted to the "Risk Component".  Although the type of evaluation is the same for the two indicators, the milestones that make up their scales are not the same. They are adapted to the component they evaluate.

Illustrations are essential in a DICRIM. However, it is necessary to pay attention to their nature. The rating given will differ

depending on what is shown in the photograph. In CI10, the maximum score is given for photographs representing the city, issues or protective measures (Table 4). They make it possible to illustrate the comments generally made in editorials which not only aim to make the reader aware but also reassure them. A photograph of damage or a phenomenon may attract the reader's attention too much. For example, they may wonder about the nature of the phenomenon represented when it is not the subject of this section. It is preferable that these photos appear later in the document, with information on the phenomenon, in

the Risk Component (Table 5). In this indicator (CI34), photographs dedicated to the phenomenon or damage.



| Name | CI34– Element represented by the photograph |
|---|---|
| Definition | If a photograph is present, this indicator evaluates what it represents. |
| Scale and references | (scale: Unacceptable 0–1, Bad 2–3, Poor 4–5, Acceptable 6–7, Good 8–10)<br><br>10: Photograph of the phenomenon and/or photo of damage;<br><br>8: Photograph of the phenomenon or damage and photographs of structural protection measures;<br><br>6: Photographs of structural protection measures;<br><br>5: Photograph of the city and/or issues;<br><br>4: Other. |
| Location characteristic | For each component dealing with a phenomenon. |

**Table 5. CI34 grid "Element represented by the photograph" for the Risk component**

Seventeen substantive indicators have been developed. Table 6 shows the example of the content indicator CI 38 "Usefulness of the data presented" dedicated to the Risk Component.

| Name | CI38 – Usefulness of the data presented |
|---|---|
| Definition | The focus is on assessing the usefulness of the component content |
| Scale and references | (scale: Unacceptable 0–1, Bad 2–3, Poor 4–5, Acceptable 6–7, Good 8–10)<br><br>10: The component contains the following elements: 1 Hazard zoning map (+ explicit legend) on 1 page and Presentation of the phenomenon and its impact on the municipality with mention of past events and Protection Measures and Alert/Rescue Organization on ½ page and Safety Instructions (Before, During and After) on ½ page.<br><br>7: The number of pages differs from one element to another within the component OR an explicit legend associated with the map is missing.<br><br>5: Lack of protection measures OR the map is missing OR the presentation of the phenomenon is present but the mention of past events in the municipality is missing.<br><br>3: The presentation of the phenomenon and its impact or safety instructions are missing, but without distinction of the phases before/during/after.<br><br>0: safety instructions are missing OR the presentation of the alert / rescue organization is missing. |
| Location characteristic | For each component dealing with a phenomenon. |

**Table 6: Grid for the content component indicator 38 "Usefulness of the data presented" for the Risk component**



This indicator combines two characteristics to be assessed: the information contained in the Risk section and the length of each of its paragraphs. Although the length of this section is specified by law (2-3 pages maximum), there is no indication as to the length of the elements it must contain. Based on expert knowledge, we have therefore specified the lengths that the different informative parts of the component must have in order for it to be as effective as possible in terms of making the information it contains understandable. To build our scale we based ourselves on a 2-page section in view of the already substantial amount of information that the DICRIM contains. Three pages per risk would risk leading to an excessive total number of pages in the document, which could discourage the reader. In addition, the DICRIM that we presented to the residents interviewed during our survey consisted of two opposing pages per risk. This feature was mostly appreciated. The scores decrease according to the length of the paragraphs or the nature of any gaps. Since the management of the reader at the time of the crisis (behavior or organization of the alert or rescue) is crucial information, the shortcomings on this subject lead to a score of 0 for this indicator.

### 3.3 Verification of suitability indicator grids

For each DICRIM, 114 indicators were used because the experts were asked to evaluate two risk components (a natural risk and a technological risk defined in the measurement protocol as the presentation of the risk headings is generally homogeneous in a given document). Three verification sessions were conducted with the same experts with improvements made to the indicators between each session.

The analysis and comparison of the experts' results and their comments highlighted the need to make modifications to some of the grids that had construction errors, scale problems or problems linked to understanding of the milestones and which complicated the evaluation. The scores obtained during the third verification were the final results on which statistical analyses were carried out. In particular, standard deviations were calculated between the scores of different assessors for the same indicator in order to test the reproducibility of the rating. We considered that a standard deviation of less than 2 between two experts for a given DICRIM was acceptable. Ninety indicators were assessed by the experts.

There were still rating gaps with a standard deviation of more than two following this third session: 25 indicators for D1; 19 for D2; and 22 for D3. Since the same indicator could be used for several different evaluations, this gives 14 indicators for D1, 3 additional indicators for D2 and 5 for D3. Of these, seven were present in the rating of the three DICRIMs. We therefore obtained 72% of acceptable indicators for D1, 79% for D2 and 76% for D3. It should be noted that no evaluator gave ratings very different from the others. These differences can be explained as follows:

- 20% (D1), 37% (D2) and 32% (D3) of these differences were due to human errors because the answers were factual and did not involve judgment (number of pages, for example).





- 36% (D1), 26% (D2) and 32% (D3) of the discrepancies were due to errors that could be attributed to the formulation of milestones that were sometimes too ambiguous or complex, a finding established on the basis of the evaluators' feedback. These milestones have been improved.

- Finally, these differences could be attributed to the subjectivity of the indicators concerned. Thus, 44% of standard deviation scores were higher than 2 for D1, 42% for D2 and 36% for D3 (11 indicators out of 90 indicators for D1, 8 for D2 and D3): for example, the indicators "Relevance of the association of text and background colors" or "Photograph resolution" which required the evaluator's personal judgment. Some people rated a resolution or contrast as satisfactory while others were more severe due to their own perception. As this was largely a visual assessment of communication, it seemed difficult to completely avoid this type of problem in the ratings. However, since the tool may be used by a non-expert in design, this type of grid nevertheless provides a framework for subjective evaluation, in order to reduce this subjectivity.

We also averaged the scores of all CIs, all SIs and the overall average CI + SI for each evaluator, i.e. the suitability of each DICRIM. Table 7 presents the averages obtained by each evaluator for DICRIM 1 (with and without taking into account subjective indicators – I-subj). There is no significant difference between the means with and without the subjective indicators, so they do not affect the final evaluation. We chose to keep them in the model because they remain relevant indicators in terms of communication suitability. Future work will need to consider clarifying and improving the milestones of these indicators in an attempt to reduce this subjectivity.

| | A1 | B1 | C1 | D1 | E1 |
|---|---|---|---|---|---|
| Average CI | 5.8 | 6.3 | 6.8 | 7.2 | 7.5 |
| Average CI (Without I-subj) | 6.6 | 6.4 | 7 | 7.25 | 7.3 |
| Average SI | 7.1 | 5.9 | 5.8 | 6 | 6.9 |
| Average SI (Without I-subj) | 7.1 | 5.9 | 5.8 | 6 | 6.9 |
| **Overall Average (*SI and CI*)** | **6.5** | **6.1** | **6.3** | **6.6** | **7.2** |
| **Overall Average (SI and CI Without I-subj)** | **6.9** | **6.1** | **6.4** | **6.6** | **7.1** |

**Table 7: Averages of the component, system and total indicators of each evaluator. The columns indicate each expert (from A to E)**





ANOVA results show that the F ratio is 0.87 and 0.58 for DICRIM 2 and 3, respectively. Since the probability value for the F test is greater than or equal to 0.05 for the 2 DICRIMs (0.4791 and 0.6739), there is no statistically significant difference between the means of the 5 assessors at the 5% significance level. On the other hand, for DICRIM 1 the ratio F is 2.79 and the probability is 0.0262. There is a statistically significant difference between the averages of the evaluators. The discrepancies

of DICRIM 1 can be explained by the fact that this DICRIM had many pages (~30), each with a large amount of information and a heterogeneous design, which could have led to rating errors by the evaluators. The results of the ANOVA, although statistically different, are therefore acceptable.

In the light of all the results, these verification sessions therefore make it possible to conclude on the robustness of the indicators developed. It should be noted that verifications and improvements to the grids are still possible.

The evaluation of the first DICRIM took on average between three quarters of an hour and one hour for all the evaluators. This time was reduced afterwards for the next DICRIM, probably due to the fact that they were familiar with the method.

## 4. Production of evaluation and improvement model

To build the model, a hierarchy was established for all the indicators using weights that were distributed for each indicator based on expert advice. Indeed, certain indicators allow the evaluation of major suitability characteristics of the DICRIM. For

example, if the visual content of a section repels the reader (too much text, too little white space, inappropriate colors, poor quality illustrations, etc.) despite the relevance of the information contained, it may not be accepted. Conversely, a visual content that makes a text more accessible may be useless if the text is incomprehensible because of its overly technical vocabulary, for example. These are therefore major elements to be taken into account when evaluating the suitability of components and the document. Other characteristics are important to consider but have a lower risk of blocking the transfer of

information (type of paper used, colors of titles, etc.). The weights selected were 0.5 (Orientation, Type of paper), 1 (*e.g.* Print format, Form of the DICRIM) and 2 (*e.g.* DICRIM format, Total number of pages, Typography). They are presented in Table 8 for the system indicators. The same procedure was carried out for the indicators of each component.

| Weight 0,5 | Weight 1 | Weight 2 |
|---|---|---|
| SI4 – Orientation | SI3 – Print format | SI1 – DICRIM format SI2 – Visual impression |
| SI5 – Type of paper | SI7 – Form of the DICRIM | SI6 – Total number of pages of the DICRIM |
| | SI11 – Multi-hazard map | SI8 – Type and Position of components in the DICRIM |
| | SI12 – Information on the insurance system | SI9 – Evaluation of added components |
| | | SI10 – Typography of DICRIM texts |
| | | SI13 – Alert in the event of a major event |

**Table 8: Weights assigned to each system indicator**





The indicators were then aggregated by weighted averages. The suitability model consists of a weighted average of the 48 CIs; a weighted average of the 13 SIs and then an overall average. Figure 4 shows the structure of this model.

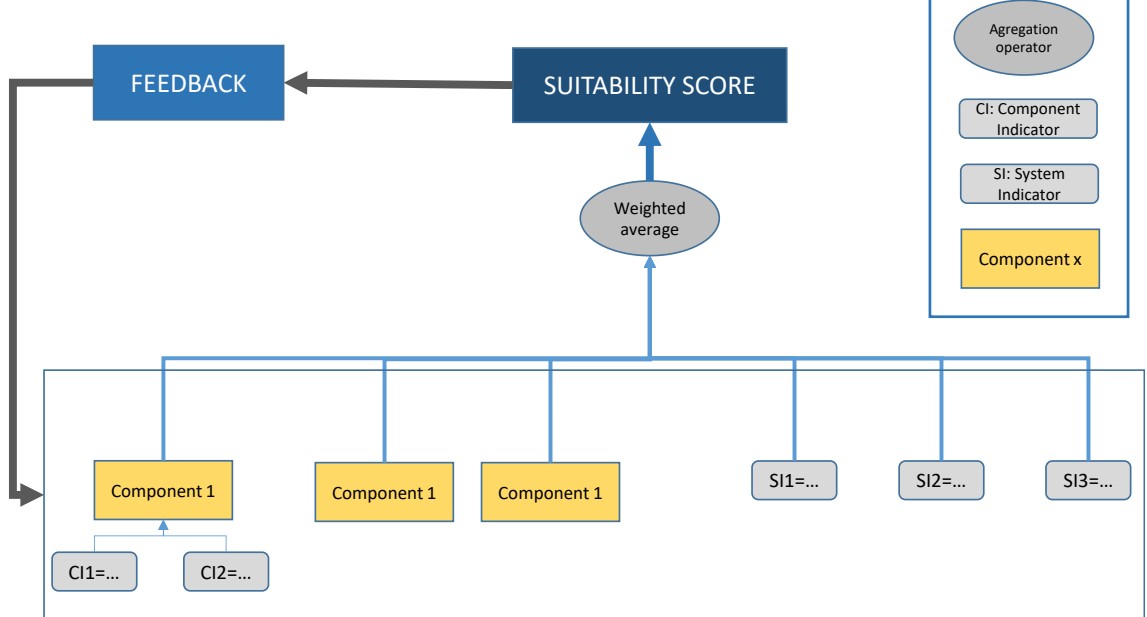

**Figure 4: Model for evaluating the suitability of the DICRIM by aggregating indicators**


We observed that the differences between the averages of each evaluator for each DICRIM also narrowed. They were finally less than 1 point apart. This shows that the gaps we observed were partly due to indicators that were less important in assessing suitability and/or that the indicators that were in good agreement were those most heavily weighted. The model allowed obtaining a quantified value of the document's overall suitability, which is interesting for communicating on the DICRIM.

While this score can provide an estimate of the overall degree of suitability of the DICRIM, this result is not sufficient, making it necessary to know what can be done to remedy an insufficient score. Indeed, the model also allows adding feedback. From a score of 0 to a score of 5 inclusive, which is the top level of the "Poor" interval of our double evaluation scale, the model proposes the actions to be implemented at the component or system level depending on the indicator concerned. These improvements will be carried out so that the indicator reaches a score of 10 for a new evaluation, ensuring better suitability of

the given component or system. For example, if System Indicator 13 "Typography of texts contained in the DICRIM" has been attributed a score of 3 because the use of bold, underlined, italic or bright colors is too frequent, the template will suggest that these graphic characteristics should be used only for words or groups of words to be emphasized.



It should also be noted that the model remains scalable and can be completed, refined and/or improved without the need for major changes in the body of the model

**Conclusion**


In this work we proposed generic decision-making tools for the municipalities responsible for implementing the DICRIM. By generic we mean that our method can be applied whatever the DICRIM considered. This is possible because the indicators stem from a functional analysis that describes the DICRIM in a conceptual way without considering particular documents (Ferrer et al., 2018). This point is very important because it permits reducing the resources to be mobilized. Municipalities will

be able to implement formalized knowledge without the need for an expert.

We made the hypothesis that the suitability evaluation process, usually made with the help of several experts, could be formalized in a decision-aid model. The formalization of the detection elements identified during a previous systemic analysis allowed the development of indicators. Their description grids were fed by several types of input data. All the indicators were verified by five experts who evaluated three existing DICRIMs in real life situations of grid use. These verifications allowed

highlighting the improvements to be made to certain indicators and then assessing the robustness of the evaluation. A model was obtained by aggregating these indicators to obtain suitability scores for the entire document and each component of a given DICRIM, regarding both the form and content of the DICRIM, and to provide feedback. The model has the advantage of offering a repeatable and reproducible structure of indicators for evaluating any DICRIM, without it being necessary to use specific resources. The suitability evaluation model also has generic components that will be easily applicable to other types

of risk communication tool. The associations and communities (Town Hall and Prefecture) we met were receptive to our approach and our results, considering them relevant.

It noteworthy that this model is the first approach and that although it provided positive results during the verification session, it should be improved in future work. Future work will need to consider clarifying and improving the milestones of some indicators in an attempt to reduce subjectivity. It would also be appropriate to propose updates, for instance the safety

instructions in the "Data utility" indicator of the Risk component, by calling on other experts such as firefighters and first-aid workers. Indeed, we esteem that some instructions are missing in the DICRIM, such as instructions for smartphone owners, since smartphones are now widely used to obtain information. It will also be interesting to conduct tests with a sample of a population thanks to a DICRIM improved by using the tools, in order to judge their relevance.

Moreover, it is important to remember that even a DICRIM improved by the models cannot guarantee good behavior on the

part of the population on the day of a major event. It does not provide a one-size-fits-all-solution but must be part of a broader preventive communication strategy. This must consist of a set of awareness-raising tools which should also be evaluated, for example, by adapting the decision-making tools we propose.



## Author contribution

Laetitia Ferrer: Conceptualization; Formal analysis; Investigation; Methodology; Validation; Writing - original draft; Writing
- review & editing

Corinne Curt: Conceptualization; Formal analysis; Funding acquisition; Investigation; Methodology; Project administration;
Supervision; Validation; Writing - original draft; Writing - review & editing

## Competing interests

The authors declare that they have no conflict of interest.

## Acknowledgments

This work was supported by the French National Research Institute for Agriculture, Food and the Environment (INRAE) and
the SUD (Provence-Alpes-Côte d'Azur) Region.

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
