# Peer review of "Evaluating the suitability of documents on the prevention of major risks intended for the general public"

_Natural Hazards and Earth System Sciences, 2021_

## Referee Comment (RC2)

[referee-annotated manuscript omitted]

---

## Author Comment (AC1)

**Article nhess-2021-192 – Ferrer et Curt**

**Responses to reviewer – RC1**

| Reviewer comment | Authors reply |
|---|---|
| The problem is connected to the topic of dissemination of risk information, and thus the question falls in the scope of the journal. | Thank you very much for your interesting comments

We hope that the changes we made following your comments will make the reading of the article more fluid |
| In fact, it's not clear the usefulness at municipal level of one synthetic index summarizing all the evaluations | As stated in the article l.405-406, "While this score can provide an estimate of the overall degree of suitability of the DICRIM, this result is not sufficient, making it necessary to know what can be done to remedy an insufficient score." The synthetic index gives an overview of the readability and legibility of the document such as it is done in the SAM method for instance but the corrective actions are triggered by the value assessed by each indicator: this was not well presented in Figure 5 and can induce a misunderstanding of the use of the synthetic index. We will therefore alter this figure to better present the process of improving the document.

We will add these elements in the new subsection 4.2 (please see below) |
| At least the perspective of introducing the index in a more comprehensive evaluation should be explicitly discussed. A section dedicated to the description of the potential use of such indicators/indexes is lacking, with a focus of different potential uses for the synthetic index or the indexes relevant to the single components. | Section 4 will be renamed: Production and use of evaluation and improvement models

We will introduce two subsections in § 4:

4.1 Determination of a synthetic index and corrective actions
This sub-section will contain the beginning of the previous Section 4 that focuses on the weighting process

4.2 Use of the models
This sub-section will be introduced as a discussion on the use of the assessment and improvement models: |

| | |
|---|---|
| The proposed approach makes use of a huge set of indicators (114) for the evaluation of a single document. Considering also that readability and legibility shouldn't be the only elements taken into account when evaluating the suitability of the document for informing the public, this high number of required evaluations could be detrimental for the use of the approach. Please discuss this in the new section (see previous comment); it could be helpful to have the indication of the mean time required for compiling the indicators for one DICRIM. | As indicated above, 112 (*not 114 - it was an error*) indicators were needed to evaluate a DICRIM considering one natural risk and one technological risk. This number may seem high, but the scoring is quick because the indicator grids guide it precisely. Moreover, for some documents, certain headings are missing altogether. In the end, for the validation sessions, the time taken by the assessors was between 45 and 60 min depending on the content of each heading.

As it is not possible to have a precise idea of the readability and legibility of the document due to their evaluation by 112 indicators, a synthetic score is relevant such as it is done in the SAM method for instance. Moreover, it is possible to provide an overall form rating and an overall content rating so that the user knows whether the readability or the legibility is the most problematic (*Figure 4/Model for evaluating the suitability of the DICRIM by aggregating indicators* will be completed by indicating a form index and a content index). If these synthetic indexes give an overview of the readability and legibility of the document, the corrective actions are necessarily triggered by the value assessed by each indicator: at the end, the user obtains a list of actions to improve the document.

The associations and communities (Town Hall and Prefecture) we met were receptive to our approach and our results, considering them relevant.

It should also be noted that the model remains scalable and can be completed, refined and/or improved without the need for major changes in the body of the model. |

[Figure]

Proposal for Figure 4

| | |
|---|---|
| The authors cite an approach already common in other contexts, without giving details about it. | More information will be given on the SAM method to clarify what are the main innovations: |
| | The SAM method is much more elaborated than readability formulas. It assesses **22** factors related […], Appropriateness. For instance, "purpose is evident", "content about behaviour", "scope is limited" and "summary or review included" refer to Content factor. Each indicator is assessed on a same scale ranging from 0 (not suitable rating) to 2 (superior rating). The final score corresponds to the sum of the value affected to the whole set of factors divided by 44 corresponding to the maximum possible total score. |
| It's highlighted that such an approach has not been applied to the risk information context, but it should be made clearer where the innovation | A summary sentence was effectively missing at the end of § 1.3. We will add the following: |

| | |
|---|---|
| stands (e.g. adaptation of the approach, identification of proper indicators, identification of sections of the document to be analyzed) | There is currently no method that combines the analysis of the form and content of a document dedicated to the communication of major risks, capable of evaluating cartographic representations and photos at two levels: the entire document and sections of documents. It is a question of adapting certain indicators existing in the literature but also of formalising its own indicators adapted to the two levels. |
| The sources for providing the definition of the indicators and defining their milestones, the authors indicate that several sources can be used. Although, a clear indication of which specific sources have been used for each indicator should be provided at least as supplement material | We will add a table providing the types of source used for each detection element. We will make this table from the detection elements rather than from the indicators because there are redundancies in the indicators. Indeed, the same type of indicator is used for several components: for example, CI10 (Table 4) and CI34 (Table 5) come from the detection element "Elements represented by the photo" (Table 3). The same sources were used for these indicators, in relation to the initial detection element. |
| Among such sources, the authors indicate a database with about fifty DICRIMs as potential sources for extrapolating the structure of the indicators [page 8, lines 207-209]. The approach is not very clear, and it can give the idea that they are based on the average scores or description of the analyzed documents. If this is the case, the specific indicator would not describe the suitability of the document, but only its ranking | Elements will be added to specify how the DICIRM database was used to build the indicators:

 This database of DICRIMs made it possible to identify concrete practices (e.g. types of photos present, colour or typography of texts, vocabulary used, etc.) serving as examples or counter-examples to describe the references associated with the possible values assigned to the indicators. For example, a census of the type of photos was made in the 50 DICRIMs for the section "Presentation of the risk" and this list was used to define the references for the indicator CI34 (Table 5): photograph of the phenomenon, damage, structural protection measures, city, issues. Thus these were used to describe the suitability of the document. |
| Still referring to the database of the DICRIMs, it's not clear if these are the documents used for the verification phase, thus not allowing a proper evaluation of the soundness of such a phase | The documents used during the validation phase were not included in the database of DICRIMs |
| When introducing the groups of experts involved in different phases, the indication of each one's years of experience is a bit funny; it could be | The reference to the experts' years of experience will be removed in favour of information on the reasons why we chose these experts: |

| | |
|---|---|
| preferable to have a general description of the criteria adopted for selecting them | Two experts in technological or natural risk analysis were more particularly involved in questions of document content and its headings, while the third as a communication expert was involved in questions of form. |
| Page 3, line 68. The title is a bit misleading, please change with something more focused on the effectiveness of communication (e.g. Effective communication in risk management) | The title is changed for: **Effective communication in risk management** |
| Page 5, line 125. Please introduce the regulatory reference defining the list of 11 mayor risks that may affect a municipality | The reference is MEDDE: Maquette nationale pour l'application du code de l'environnement - Articles L125-2 et R125-5 à R125-27, Paris, France, 2013 – It is cited in the revised version |
| Page 6, Figure 3. The picture in a bit unclear. Shouldn't the definition of corrective actions follow a step of evaluation of the indicators for the specific DICRIM? Please clarify and/or modify the picture | Figure 3 presents the methodology adopted. It focuses on (i) the indicator construction and choice of aggregation rules and (ii) the definition of corrective actions to improve DICRIMs. The use of the indicators is shown in Figure 5.

To clarify Figure 3, changes will be made:
- "aggregations" replaced by "choice of aggregations"
- "definition of corrective actions" by "Definition of corrective actions to improve the DICRIM"
- the box "DICRIM improvement" will be deleted |
| Pages 13 and 14, tables 4 and 5. The two grids refer to the same detection element, but there's no reference to the specific component they are linked to; adding the component could help the reader in interpretating the table | Changes will be done: the components are now more clearly identified
Table 4: *Cp2* - At the beginning of the DICRIM – on the page where the editorial is located
Table 5 (and Table 6): *Cp5* - For each component dealing with a phenomenon |
| Page 13, table 4. Please clarify "form 10" in the caption | The table title will be changed for: Grid of the form indicator 10 "Element represented by the photograph" for the Editorial component |

---

## Author Comment (AC2)

**Article nhess-2021-192 – Ferrer et Curt**

**Responses to reviewer – RC2**

| Reviewer comment | Authors reply |
|---|---|
| I encountered difficulties in reading the text, since it is not fluid, and in the understanding the workflow you used to perform your analysis. Some parts of the manuscript are, in my opinion, to be presented differently. | Thank you very much for your interesting comments.
We hope that the changes we made following your comments will make the reading of the article more fluid |
| First of all, I did not understand how figure 3 can reflect the method used. If it is clear that a Systemic Analysis is performed for the suitability detection elements, but it is not clear to me how all the other sources used to select and identify other indicators were processed (the orange ellipses in figure 3). For example, the results of the questionnaire survey dedicated to the design of the cover page were used for the SI or for the CI indicator? I think you have used for the component analysis since the cover page is one of the document components but it is not clear when reading the manuscript. And it is the same for what you named "DICRIM database" or for the "opinions of residents". | Indeed, the figure does not quite present the process that was used: first a systemic analysis to define the detection elements, then the use of 5 different sources (literature, DICRIM database...) to go from detection elements to indicators. We therefore modified the figure to show this process

At the beginning of § 2.5 we will add a sentence: Detection elements are formalized as indicators. |

| | The different sources are described l. 192-219. We hope that the changes made to Figure 3 will provide a better understanding of the relevance of the different sources |
|---|---|
| A second problem concerns the experts: did they provide indicators or they were included in the verification phase only, or both? | Two groups of expert were involved: a first one (3 experts) worked on the formalization phase and a second one (5 experts) performed the validation tests. We will specify in the article that 2 independent groups of experts have worked |
| I have also encountered a problem with some terminology, it is my fault, but it is not clear to me if anchoring points on the scales, references and milestones can be considered the same concept. Please clarify better the differences. | You are absolutely right. We used several terms for the same thing and this is confusing. We will adopt a single term in the revised paper |
| Regarding the scales of analysis, it is clear the reasons for using more than one. Conversely, I found difficulties in the understanding the way you have used the indicators for the two scale of analysis. | The assessment is carried out using the quantitative scale (0-10) but we introduced a qualitative scale (unacceptable – good) because experts look first for the qualitative class (e.g. "Poor") and then for the more precise score (e.g. "4") |
| Comments from Supplement document | |
| Please, specify what are you meaning with "fields it contains". | We will give examples the fields contained in DICRIMs:

System Indicators for the entire document and Substantive and Formal Component Indicators for the fields it contains (e.g. editorial from the mayor, DICRIM presentation, substantial information for each hazard affecting the town, emergency phone numbers) |
| it is merely a list. it could be more interesting to add few details. | We will add some words to describe each initiative |
| shorter is better, deleting the first part, it sounds better: suitability of risk management materials | The title will be shortened |
| Please, specify what kind of model you are meaning. Document model? | We refered here to the French National Model for the Application of the Environment Code.

We will change "this model" by "this National Model" |

| | |
|---|---|
| composed of: (i) an editorial from the mayor, (ii) ...... (iii)...... | We will adopt this presentation |
| Please add (SIx) and the number of this type of indicator | SIx and the number of SIx have been added in the new version of Figure 3 |
| Please add (CIx) and the number of this type of indicator | CIx and the number of CIx have been added in the new version of Figure 3 |
| It could be helpful for understanding the work flow, to add close by the arrows, the type of analysis carried on | The type of analysis carried on has be added close to the arrows (new version of Figure 3) |
| Is it the same of the grey System Analysis in Figure3? | Yes, it is, it corresponds to the grey System Analysis in Figure 3. We will precise this : (stemming from the systemic analysis (Ferrer et al., 2018) **– *Figure 3***) |
| Please try to explain in more detail | We will give more details to explain why the various elements of description (definition, measurement scale, anchorage points…) are important to formalize indicators. We will more explain the anchorage points as references because in the grids we use the term "references" (in relation with your comment "3;5,10 - are these numbers references?") |
| I have a question: on what basis did you set the scores? why does the A4 format have 5 as a score? | The scores are based on the different types of source. For instance, residents indicated that they preferred the A5 format than the A4 format, so we assigned the score 10 to the A5 format and 5 to the A4 format |
| 3;5,10 - are these numbers references? | These numbers are references – we will more explain the role of anchorage points and references in the text above Table 1 |
| Please try to be more clear. It is an important point in the understanding the following steps of the method. The anchoring points are the same of milestome? | We effectively used different terms to describe the same thing. We will homogenise the various terms in the article for the sake of clarity |
| yes, you are right, but how do you use all the five sources? | As also requested by Reviewer 1, we will introduce a table presenting which source(s) is(are) used for which detection elements |
| CI1 is the only indicator extrapolated from the survey? | CI1 is the only indicator extrapolated from the survey |
| is it a verificatioin grid? | We will change the title by Verification phase |

| | |
|---|---|
| Is the formulation of the milestones subject to a subjective evaluation or did it arise from a procedure? | The milestones (references on the assessment scale for each indicator grid) were determined by the procedure described in § 2.5 (using several types of source) |
| In my opinion the specification of the number of years of experience is not important to say, here and in the other sentences of the manuscript. you decided he/she is an expert and the number of year is not an experience indicators, or not the only one!!! | As also requested by Reviewer 1, the reference to the experts' years of experience will be removed in favour of information on the reasons why we chose these experts:

Two experts in technological or natural risk analysis were more particularly involved in questions of document content and its headings, while the third as a communication expert was involved in questions of form. |
| I'm not sure to understand the meaning. did they test the documents during an emergency? | "in real life conditions" will be removed. This does not add anything special and is confusing. |
| the exact name is relevance or pertinence (as in table 3) | We will change "Pertinence" by "Relevance" in Table 3 |
| Please, explain the concept in a better way | The sentence refers to the verification phase and will be changed, completed and moved to the beginning of § 3.3:

Assessing all the risk headings of a DICRIM would lead to the use of too many indicators. For each DICRIM, 112 (*not 114 - it was an error*) indicators were used because the experts were asked to evaluate two risk components (a natural risk and a technological risk defined in the measurement protocol as the presentation of the risk headings is generally homogeneous in a given document). |
| how do ou get the 114 indicators. Which indicators do they include among those described so far?
114 indicators including: 48 CI, 13 SI..... | We will simplify the confusing text. We have defined 48 types of indicator, some of which are broken down into several components (e.g. the colour of titles). In total, we have 112 (*not 114*) indicators: 99 CI and 13 SI. The text concerned will be altered throughout the article (§ 3.1, § 3.3 and § 4) to keep only the number of 112 indicators. |
| Please, specify the reason for which you use three verification sessions. | We will add a paragraph to explain why we led three verification sessions: add at the end of l.337)

The necessary changes were made after the first session. A further session (session 2) was conducted with the experts to check that the changes were appropriate. An improvement was achieved but some further errors in scoring or lack of clarity in the definition of |

| | indicators were still noted. We therefore modified the indicator grids and conducted a third and final session to check the improvement of the grids (no more questions from the experts and less discrepancies in scoring between them) |
|---|---|
| Please, try to explain why the 114 indicators reduced to 90. | We will complete the sentence "Ninety indicators were assessed by the experts": Indeed, not all indicators are evaluated for the 3 DICRIMs because some DICRIMs did not contain a component: for example, DICRIM1 did not include the municipal poster. Thus, the number of indicators evaluated on the 3 DICRIMs was 90. |
| where can I see your results? | We can add an appendix with 3 tables (one per DICRIM) presenting the assessment of the 112 indicators by the 5 experts |
| Maybe, you could first divided in factual indicators from those subjective. | The first two items in the list concern factual indicators while the third concerns subjective indicators |
| As an example ?? why only the dicrim 1 avarages? | The results for DICRIM 1 are presented in Table 7 as an example. We can add the results for DICRIMs 2 and 3 directly in the article or in the appendix |
| the weights you use come from literature or are based on your experience? please specify | These weights come from our experience |
| But you have results for three of them | We will add: "We have shown its feasibility by applying it to three examples" |
| which, however, it would be better to consult. | We can add: "which, however, could be consulted." |